# Assessment of Patients’ Quality of Care in Healthcare Systems: A Comprehensive Narrative Literature Review

**DOI:** 10.3390/healthcare13141714

**Published:** 2025-07-16

**Authors:** Yisel Mi Guzmán-Leguel, Simón Quetzalcoatl Rodríguez-Lara

**Affiliations:** School of Medicine, Universidad Autónoma de Guadalajara, Zapopan 45129, Jalisco, Mexico; yisel.guzman@edu.uag.mx

**Keywords:** quality of care, quality healthcare improvement, quality models, strategies of improvement

## Abstract

**Introduction:** Assessing the quality of patient care within healthcare systems remains a multifaceted challenge due to varying definitions of “quality” and the complexity of care delivery structures worldwide. Patient-centeredness, institutional responsiveness, and contextual adaptability are increasingly recognized as core pillars in quality assessment. Objective: This narrative literature review aims to explore conceptual models and practical frameworks for evaluating healthcare quality, emphasizing tools that integrate technical, functional, and emotional dimensions and proposing a comprehensive model adaptable to diverse health system contexts. **Methodology:** A systematic literature search was conducted in the PubMed, Scopus, and Cochrane Library databases, covering the years 2000 to 2024. Studies were selected based on relevance to quality assessment models, patient satisfaction, accreditation, and strategic improvement methodologies. The review followed a thematic synthesis approach, integrating structural, process-based, and outcome-driven perspectives. **Results:** Core frameworks such as Donabedian’s model and balancing measures were reviewed alongside evaluation tools like the Dutch Consumer Quality Index, SERVQUAL, and Importance–Performance Analysis (IPA). These models revealed significant gaps between patient expectations and actual service delivery, especially in functional and emotional quality dimensions. This review also identified limitations related to contextual generalizability and bias. A novel integrative model is proposed, emphasizing the dynamic interaction between institutional structure, clinical processes, and patient experience. **Conclusions:** High-quality healthcare demands a multidimensional approach. Integrating conceptual frameworks with context-sensitive strategies enables healthcare systems to align technical performance with patient-centered outcomes. The proposed model offers a foundation for future empirical validation, particularly in resource-limited or hybrid settings.

## 1. Introduction

Assessing the quality of patient care within healthcare systems remains a complex endeavor due to the lack of a universally accepted definition and the multifaceted nature of care delivery [1,2]. Although quality is broadly associated with excellence, the absence of defects, and the achievement of intended outcomes, in healthcare, it requires a more nuanced and contextual understanding [1,3]. The Institute of Medicine defines quality of care as “the degree to which health services for individuals and populations increase the likelihood of de-sired health outcomes and are consistent with current professional knowledge” [4]. However, the operationalization of this definition remains challenging across diverse healthcare contexts.

Various conceptual models have been proposed to evaluate and enhance quality in healthcare, including the Donabedian model, which categorizes quality into three interrelated components: structure, process, and outcomes [5,6,7]. Despite its widespread adoption, this model alone is insufficient to fully capture the patient’s subjective experience or the dynamic interactions within modern health systems. Furthermore, no single objective scale or framework is universally applicable, resulting in the proliferation of context-specific tools and methodologies [8]. Among these, the concept of person-centered care (PCC) has emerged as a pivotal paradigm for quality assessment [9,10,11,12,13]. PCC emphasizes individualized care tailored to patients’ preferences, values, and cultural backgrounds. It also promotes shared decision-making, empathy, compassion, and respect—qualities often underrepresented in traditional technical evaluations [9,10,11,12,13]. Therefore, integrating PCC into quality frameworks ensures a more holistic and humane approach to evaluating healthcare systems.

In this context, tools such as SERVQUAL and Importance–Performance Analysis (IPA) have gained relevance [14,15,16,17,18]. SERVQUAL measures perceived service quality across five dimensions: tangibility, reliability, responsiveness, assurance, and empathy [14,15]. IPA complements this by plotting the importance of each attribute against its actual performance, helping prioritize interventions [16,17,18]. When applied jointly, these tools offer valuable insights into the alignment between patient expectations and service delivery, facilitating targeted quality improvement strategies [14].

This narrative review aims to analyze the conceptual and practical frameworks used to assess quality of care in healthcare systems [19]. Specifically, it examines models that integrate patient satisfaction, system performance, and person-centered care. By synthesizing the international literature and identifying methodological trends and gaps, this review seeks to contribute to the development of adaptable and comprehensive strategies for improving healthcare quality globally.

## 2. Methodology

This narrative literature review employed a structured yet conceptually flexible approach, allowing for the integration of both qualitative and quantitative findings across diverse healthcare settings. Unlike systematic reviews, which are focused on exhaustive and replicable searches, narrative reviews aim to synthesize existing knowledge, identify conceptual frameworks, and offer interpretive insights that connect evidence across contexts.

The literature search was conducted across four primary databases: PubMed, Scopus, Web of Science, and Google Scholar. Boolean operators and controlled vocabulary (MeSH terms) were used to enhance precision. The search strategy combined terms such as “quality of care”, “healthcare quality assessment”, “patient satisfaction”, “person-centered care”, “SERVQUAL model”, and “importance-performance analysis (IPA)”, covering publications from January 2000 to December 2024 (Table 1). Only peer-reviewed articles in English were considered.

The selection criteria included studies that (1) addressed the quality of care from a patient-centered or service-delivery perspective, (2) utilized conceptual models such as Donabedian, SERVQUAL, IPA, or PCC, and (3) were conducted in institutional or systemic healthcare settings. Exclusion criteria encompassed editorials, opinion pieces, dissertations, gray literature, and studies focused solely on economic or managerial outcomes without integrating patient experience. A total of approximately 85 articles were initially screened, with 42 meeting the inclusion criteria after full-text review.

To reduce selection bias, two authors independently screened titles, abstracts, and full texts, resolving discrepancies through discussion and consensus. Although formal risk-of-bias tools were not applied due to the narrative nature of this review, the inclusion of studies from diverse countries and settings aimed to ensure representativeness and reduce publication bias. Studies were categorized thematically according to methodological design, healthcare context (public vs. private; inpatient vs. outpatient), and region (high-, middle-, and low-income countries).

A thematic framework analysis was employed to extract core constructs related to service quality models, dimensions of care, patient-reported gaps, and accreditation strategies. An emphasis was placed on identifying both converging trends and areas of divergence. The methodological logic of this review aligns with previous narrative syntheses applied to the evaluation of health system performance (e.g., [8]).

This article adopts a narrative review design, which prioritizes conceptual exploration, the synthesis of key themes, and interpretive discussion over standardized outcome reporting. In contrast to systematic reviews, narrative reviews do not require a separate “Results” section. Instead, the analysis is presented in the form of structured thematic discussions that reflect the authors’ synthesis of relevant literature. This methodological approach is consistent with international quality standards for narrative reviews, such as the SANRA scale [20], which emphasize clarity, logical argumentation, and interpretive depth over formalized data extraction and reporting.

## 3. The Donabedian Model and the Importance of Equilibrium in Quality Assessment

One of the most influential frameworks for assessing healthcare quality is the Donabedian model, which organizes quality into three interconnected dimensions: structure, process, and outcome [3,4,5,6,7,21]. This model has been extensively applied and cited due to its clarity, applicability, and conceptual elegance. “Structure” refers to the attributes of the settings in which care occurs, including facilities, equipment, and human resources. “Process” concerns the methods, interactions, and activities through which care is delivered. “Outcome” denotes the effects of healthcare on the health status of individuals and populations.

The Donabedian model is grounded in the assumption that improvements in structure will lead to enhanced processes, which in turn yield better outcomes. However, this linear causality does not always hold in practice. For example, the presence of advanced infrastructure (structure) does not guarantee high-quality interactions (process), nor do excellent clinical procedures always lead to optimal patient recovery (outcome) [3,4,5,6,7,21]. Hence, a more dynamic and integrative interpretation of the model has gained traction in the recent literature. This interpretive shift introduces a fourth conceptual dimension: the balance or equilibrium among structure, process, and outcome. Rather than evaluating these components in isolation, this approach emphasizes their interdependence and the need to optimize their alignment. Equilibrium in this context refers to the continuous adjustment and calibration of each dimension in relation to the others, ensuring that investments in structure translate into meaningful processes, and that both contribute to improved outcomes [2,5,22,23].

Several scholars argue that without balance, the model risks fragmentation or misalignment. For instance, an overemphasis on outcomes without attending to processes can result in superficial assessments or neglect of patient experiences. Similarly, focusing heavily on structure can lead to resource-intensive systems that fail to deliver value. Therefore, balance serves as a moderating principle that enhances the interpretive power and practical application of the Donabedian framework [2,5,22,23]. This integrated perspective is especially relevant in the current landscape of healthcare reform, where systems are increasingly tasked with delivering value-based care. It also resonates with the principles of person-centered care, as equilibrium encourages attention to not only what care is delivered but also how and under what conditions.

A study conducted in the Netherlands exemplifies the importance of these interrelated elements. Researchers analyzed survey data from patients undergoing different procedures, including hip or knee replacement, cataract surgery, treatment for varicose veins, spinal disc herniation, and rheumatoid arthritis [24]. They employed instruments such as the Dillman method, the U.S.-based Consumer Assessment of Healthcare Providers and Systems (CAHPS), and the Dutch Consumer Quality Index (CQ-index), both of which incorporate questions related to structure, process, and outcomes.

The study’s regression analysis revealed that process-related factors, particularly communication and shared decision-making, accounted for the largest proportion of variance in patients’ overall care ratings. Structural factors, including waiting times and care coordination, also played a significant role, especially in surgical patients. Interestingly, outcome-related experiences, such as improved physical functioning, had a relatively minor impact on the overall patient ratings.

These findings reinforce the notion that healthcare quality should not be evaluated based on a single element. Institutions striving for high-quality care must consider the dynamic interaction among structure, process, and outcome and, when applicable, include balancing measures. Ultimately, achieving meaningful improvements requires tailored strategies aligned with the needs of the target population and a balanced approach that considers the trade-offs between different quality elements. From a person-centered care perspective, structure should facilitate empathetic and effective provider–patient interactions, not just provide physical resources. Processes must go beyond clinical guidelines and incorporate communication, emotional support, and shared decision-making. Outcomes should reflect not only clinical success but also the patient’s experience and satisfaction.

The Donabedian model, when expanded to consider equilibrium, becomes particularly useful in the evaluation of healthcare service quality as perceived by patients. It enables the identification of systemic weaknesses that affect patient satisfaction and helps determine whether deficits lie in structural resources, process execution, or the effectiveness of care outcomes.

In summary, the Donabedian model remains foundational for healthcare quality assessment. However, its full utility is realized when structure, process, and outcome are interpreted not just as discrete elements but as components of a balanced system. Recognizing and operationalizing this equilibrium is essential for designing and evaluating interventions that genuinely enhance the quality of care across diverse contexts.

## 4. Quality Models and Dimensions in Healthcare Systems

The assessment of healthcare quality has evolved over the decades, guided by the need to provide not only clinically effective but also patient-centered services. Service quality in healthcare is typically composed of two main components: technical quality and functional quality [25,26]. Technical quality refers to the clinical outcomes of care, while functional quality encompasses the internal processes involved in delivering that care. Several conceptual models have emerged to frame and evaluate these dimensions, offering structured lenses to analyze how healthcare services meet patient needs, ensure safety, promote equity, and deliver value-based care.

Among the most widely recognized frameworks is the World Health Organization (WHO) model, which outlines six core dimensions of quality: effectiveness, efficiency, accessibility, acceptability/patient-centeredness, equity, and safety (Table 2) [27,28,29]. This model is particularly valuable for guiding policy-level decisions, benchmarking system-wide performance, and framing global health initiatives. Its strengths lie in its comprehensiveness and alignment with health system goals, although its broad scope can sometimes limit its specificity in micro-level evaluations [27,28,29].

Another extensively used framework is the SERVQUAL model, which originated in the service industry but has been adapted to the healthcare context to measure patient perceptions across five dimensions: (I) tangibility—physical facilities, staff appearance, and equipment; (II) reliability—accurate service delivery; (III) responsibility—willingness to assist patients; (IV) assurance—confidence and knowledge demonstrated by staff; and (V) empathy—individualized attention (Table 2). SERVQUAL emphasizes the gap between expected and perceived service, making it useful for identifying quality shortfalls from the patient’s perspective [14,15]. Its strength lies in capturing subjective patient experience, but it may overlook objective clinical outcomes and is sensitive to cultural and contextual interpretations. This model is widely used in countries such as Turkey, Iran, Saudi Arabia, and Romania [15]. It evaluates functional quality through its five core dimensions and uses 44 questions divided equally between expectations and perceptions. Despite its wide use, the model has limitations, including susceptibility to bias in small samples and challenges in translating the survey into other languages. For instance, in Iran, results showed dissatisfaction across all five dimensions, particularly in responsibility and reliability, while tangibility and empathy exhibited the smallest gaps between expectations and perceptions. These findings suggest a misalignment between what patients expect and the care they perceive, potentially stemming from inadequate staff attention or poor survey adaptation. To improve functional quality, recommended strategies include supervisor feedback mechanisms and the employment of skilled translators for accurate cultural contextualization [15].

The Importance–Performance Analysis (IPA) technique complements SERVQUAL by mapping performance scores against the importance assigned by patients or stakeholders to different service elements [16,18]. The resulting matrix highlights four action areas: “Keep up the good work”, “Concentrate here”, “Low priority”, and “Possible overkill”. This allows decision-makers to strategically allocate resources where they matter most. IPA enhances decision-making but depends heavily on the accuracy and consistency of respondents’ perceptions [14,16,17].

A clinical example of IPA application can be found at the Kerman Medical Sciences University in Iran, where it was used to assess the quality of inpatient services in teaching hospitals [17]. Eight dimensions were evaluated: tangibility, reliability, empathy, service delivery, social accountability, service organization, responsiveness, and assurance (Table 2). Patients rated each dimension using a five-point Likert scale, and the results were visualized using a four-quadrant IPA matrix. The study found significant gaps between expectations and perceptions across most dimensions. Assurance received the highest perceived performance ratings, while social accountability was rated the lowest in both expectations and perceptions [17]. These insights revealed that although patients had confidence in the clinical competency of hospitals, they felt underserved in areas linked to community responsibility (Figure 1). Recommendations from the study included assigning dedicated personnel for patient education in smaller group settings, reducing clinical overload, and allocating specific staff for patient care to improve individualized attention.

This figure depicts a concentric conceptual model centered on the core element “Healthcare Population Needs”, which is surrounded by two interrelated dimensions: responsiveness and social accountability. These components are represented as semicircular domains encircling the central concept, symbolizing their shared focus on improving health system alignment with community expectations and demands. From the responsiveness domain, three key operational elements emerge: (1) the development of specific programs targeted at the most prevalent pathologies within the population; (2) the implementation of actions informed by citizen report cards, including public health campaigns, the involvement of social committees, and outreach through social media; and (3) the formulation of management protocols and contingency plans to address emerging needs and health crises. Conversely, the social accountability domain comprises four interconnected elements: (1) the design of programs focused on addressing social problems at local, regional, and national levels; (2) the use of social research methods—including campaigns, committee involvement, and media promotion—to understand public concerns; (3) the utilization of epidemiological data to guide planning; and (4) the integration of community feedback mechanisms, such as report cards, complaint boxes, and satisfaction surveys. The figure underscores the structural and functional linkage between these two dimensions and emphasizes their shared purpose in guiding healthcare systems to become more inclusive, responsive, and aligned with the evolving needs of the populations they serve.

Across these models and studies, patients consistently expect individualized attention and well-equipped, clean facilities. Improvement is needed in social accountability, responsibility (Figure 1), and reliability, highlighting perceived shortcomings in staff availability, willingness to help, and consistency in service delivery. Among these, reliability stands out as a dimension that is closely linked to institutional infrastructure and workforce capacity. Addressing it can significantly enhance care quality (Figure 2).

This figure presents a hierarchical conceptual map outlining the key elements that contribute to reliability in healthcare service delivery. At the center of the model is the concept of reliability, which encompasses both institutional infrastructure and personnel-related factors. On the left side, the dimension of “Granting” is represented, including the provision of correct patient interaction, diagnosis, procedures, and treatment. These processes are supported by elements such as knowledge of organizational procedures, soft skills, professionalism, medical knowledge, healthcare equipment, and adherence to safety protocols. On the right side, the concept of “Providing Accurate Service” is detailed, emphasizing the distribution and organization of healthcare personnel. This includes the assignment of staff exclusively to student education—favoring smaller groups and reduced patient loads—as well as personnel focused solely on patient care, supported by administrative staff, document digitalization, and artificial intelligence tools. Additionally, the model highlights the importance of staff education, direct feedback mechanisms, and continuous professional development in ensuring service accuracy and reliability. Overall, the map illustrates the interdependent relationships among organizational resources and human factors that influence the consistent delivery of high-quality care. Institutions are responsible for investigating, identifying, and addressing deficits in social accountability and responsiveness as part of their broader mission to promote health awareness and improve healthcare services.

When SERVQUAL and IPA are used in combination (Table 3), they offer a powerful dual lens: SERVQUAL identifies perceived gaps in service quality, and IPA helps prioritize interventions based on what patients value most [14,16]. This synergy has been applied successfully in diverse healthcare settings, including outpatient clinics, surgical departments, and intensive care units. However, this combined approach also inherits limitations such as reliance on patient memory, response bias, and the absence of clinical outcome indicators.

Expanding on these models’ applications, it is essential to understand their relevance in patient-centered care. Quality should not be defined solely by institutional benchmarks or technical success but should also account for the patient’s voice, expectations, and lived experience. In this regard, models that integrate subjective and objective indicators offer a more holistic assessment. Integrating these models into a healthcare quality strategy requires contextual sensitivity. The WHO model may guide national health reforms or accreditation criteria; SERVQUAL is best suited for evaluating interpersonal dynamics and service delivery; IPA helps institutions identify operational priorities and align quality improvement initiatives with user expectations. A critical insight is that no single model is universally sufficient; rather, combining models enables a multi-dimensional view that bridges system-level objectives with individual patient experiences.

### Patient-Centered Dimensions of Quality

In addition to technical efficiency and clinical outcomes, the quality of healthcare must also be evaluated through dimensions that reflect the human experience of care. This perspective is central to the framework of PPC, which emphasizes the importance of tailoring care to the individual’s values, needs, preferences, and life context [9,10,11,12,13,30]. Key dimensions of PCC include empathy, which involves recognizing and understanding patients’ emotions and personal experiences; responsiveness, which is defined as the healthcare system’s ability to adapt to specific patient expectations and conditions in real time; respect and compassion, which refer to treating all individuals with dignity, kindness, and attentiveness; and involvement, which entails actively including patients and their families in decision-making processes about their care. These principles are supported by empirical and conceptual models such as the one proposed by Santana et al. (2018) [9], which operationalizes PCC into healthcare evaluation and policy, and the Picker Principles, which highlight essential aspects of quality, including emotional support, continuity, and the inclusion of family and community. Integrating these dimensions into quality assessments fosters more humane and ethical care and has been associated with improved clinical outcomes, greater patient satisfaction, and stronger trust in healthcare systems over time.

Although the comparative table outlines the core features of the Donabedian, SERVQUAL, and PCC models (Table 3), their practical utility largely depends on the specific goals of quality assessment. Table 3 highlights how each model addresses key quality dimensions, offering a synthesized view that supports the integrative framework proposed in this review. For institutions that prioritize structural reform and measurable process improvements, the Donabedian model provides a logical starting point. Conversely, when patient perception and experience are central to evaluation, SERVQUAL offers a nuanced lens—albeit one that must be adapted to clinical realities. In settings that strive for holistic and participatory care, PCC frameworks are best suited, although they demand more profound organizational shifts. Thus, a layered or hybrid approach—where structural–process–outcome metrics are complemented by experience-based data—may offer the most comprehensive and patient-centered method for improving healthcare quality across diverse systems.

## 5. Strategies for Improving Quality in Healthcare Systems

Improving quality in healthcare systems requires the implementation of effective Quality Improvement Strategies (QIS). According to Scott’s review, “What are the most effective strategies for improving quality and safety of health care?”, several approaches have been categorized based on their relative effectiveness [31]. The most impactful strategies include (I) Clinical Decision Support Systems (CDSS), which provide healthcare professionals with knowledge and patient-specific information to enhance decision-making; (II) Clinical Practice Guidelines, which offer systematically developed recommendations for specific clinical situations; (III) audit and feedback mechanisms that compare current performance against standards to guide improvement; (IV) patient-mediated quality improvement strategies, which involve patients in their own care processes; chronic disease management programs for long-term illness management; and specialty outreach programs focused on enhancing care in specific clinical areas [31].

### 5.1. Clinical Decision Support System and Clinical Practice Guidelines

Scott’s review of high-quality randomized trials identified key factors that contribute to the success of CDSS implementation. These include the integration of decision support at the time and location of clinical decision-making, automatic provision as part of the workflow, the delivery of actionable recommendations rather than just evidence tables, and the use of computerized systems [31].

Similarly, Clinical Practice Guidelines are more effective when they are tailored to local needs, supported by active educational interventions, and reinforced with patient-specific reminders [31,32,33]. Although these strategies are often provided at the national or international levels by organizations such as the World Health Organization, the Centers for Disease Control and Prevention, and the Food and Drug Administration, they are most effective when they are implemented locally within healthcare institutions through internal consensus, adapting to contextual realities while still adhering to global standards.

### 5.2. Audit and Feedback: Patient-Mediated Quality Improvement Strategies

Audit and feedback mechanisms have proven especially effective in enhancing test ordering and preventive measures, particularly when initial adherence to guidelines is low or when feedback is intensive and detailed [34,35,36].

Patient-mediated quality improvement strategies are most successful when incorporating self-monitoring tools, patient self-management programs, motivational initiatives, and multifaceted interventions [31]. These strategies are bidirectional, involving both healthcare professionals and patients. Their success largely depends on mutual receptiveness: healthcare professionals must be open to feedback and well-trained, while patients must be encouraged to adhere to treatments and report adverse effects. Feedback directed at healthcare professionals is typically guided by governmental agencies, while feedback to patients depends on the ability and training of healthcare staff [36].

Overall, these strategies improve adherence, encourage transparency, and empower patients to participate actively in their care.

### 5.3. Chronic Disease Management and Specialty Outreach Programs

Chronic disease management programs are designed to support patients in managing long-term illnesses, improving treatment adherence, alleviating symptoms, preventing disease progression, and enhancing quality of life [30,37,38,39].

A review of 21 studies highlighted the significant benefits of these programs in managing conditions such as chronic obstructive pulmonary disease, ischemic heart disease, congestive heart failure, and diabetes mellitus [31]. The effective implementation of disease management programs can empower patients by helping them identify the most burdensome and persistent symptoms, thereby fostering a holistic understanding and control of their condition.

These programs often include continuous patient education, enabling individuals to track their progress and recognize improvements over time [39]. The incorporation of such patient-centered strategies reflects the direct link between healthcare system quality and patients’ lived experiences, ultimately contributing to more resilient and responsive healthcare delivery.

### 5.4. PCC-Based Strategies

Several quality improvement interventions embody the principles of PCC by fostering active patient participation, emotional connection, and responsiveness to individual needs [9,10,11,12,13]. Among these, patient self-management programs are particularly effective in enhancing autonomy and confidence by equipping individuals with the knowledge and tools to take an active role in managing their health conditions. Motivational interviewing, a communication technique grounded in empathy and collaboration, strengthens the therapeutic relationship by building trust and encouraging treatment adherence. Shared decision-making further aligns healthcare delivery with patient values and preferences, allowing patients to participate meaningfully in choosing among diagnostic and treatment options. In parallel, patient feedback systems, such as satisfaction surveys or suggestion mechanisms, offer continuous insight into patient perceptions and needs. These systems create opportunities for healthcare teams to adapt and refine care processes in real time. Collectively, these PCC-based strategies promote bidirectional communication, reinforce mutual respect between patients and providers, and enable healthcare systems to evolve in ways that are both person-centered and outcome-oriented. Their integration contributes not only to greater satisfaction and trust but also to more efficient, individualized, and sustainable models of care [9,10,11,12,13].

## 6. Certification and Accreditation Models

Once a healthcare institution becomes operational, its various components should seek certification, followed by an evaluation and potential accreditation of the institution as a whole [37,40]. Certification is defined as a process in which an external organization evaluates whether an individual, service, or system meets predefined, specific standards [41]. It typically focuses on specialized aspects of a healthcare process, such as quality in specific services, safety protocols, or the use of technology.

In contrast, accreditation is a broader process through which an external entity evaluates an entire healthcare organization—such as a hospital or clinic—to determine whether it meets institutional-level standards for quality and continuous improvement (Table 4). Accreditation considers not only specific clinical processes but also the overall organizational performance in areas such as patient safety, care quality, governance, and leadership [41]. As part of the description of the quality assessment, it is necessary to describe the organizations and boards [40,42,43,44].

### 6.1. International Organization for Standardization (ISO) 9001 Certification

The ISO 9001 certification program is a globally recognized quality management system used across sectors, including healthcare, to ensure consistent quality improvement and operational efficiency [45]. This standard outlines general requirements for organizational structure, process management, customer satisfaction, continuous improvement, and the effectiveness of quality systems (Figure 3). ISO 9001 certification evaluates whether an organization can consistently deliver services that meet both customer and regulatory standards [45].

This figure illustrates a triangular conceptual model that outlines three fundamental domains involved in the comprehensive assessment of healthcare organizations. At the center lies the unifying concept “Assessment of Healthcare Organizations”, which represents the ongoing and dynamic process through which institutions evaluate their operational performance, identify strengths and weaknesses, and implement strategies for continuous improvement. Each vertex of the triangle corresponds to a critical dimension of institutional evaluation. The upper corner represents functions and systems of patient care, which encompass essential components such as access to care, patient assessment, clinical care delivery, and education of both patients and their families. The right corner is dedicated to competence, defined as the combination of knowledge, skills, experience, and behaviors demonstrated by healthcare personnel—elements that directly influence the quality and safety of patient care. The left corner focuses on the operation and management of the organization, including leadership effectiveness, information management, infection control practices, and engagement with the physical infrastructure. This triangular framework emphasizes that the evaluation of healthcare organizations must be holistic, continuous, and interdependent. By systematically assessing these three domains, institutions can more effectively address performance gaps, reinforce areas of strength, and strategically evolve the quality of their care delivery systems.

This certification gained further significance due to its alignment with national internal control regulations in several countries, highlighting its relevance in regulatory frameworks. ISO 9001 certification confirms adherence to internationally accepted quality management standards and reflects an institution’s commitment to quality enhancement and accountability.

### 6.2. Joint Commission International

The Joint Commission International (JCI) is a global accreditation program specifically developed for healthcare organizations, particularly hospitals [37]. Its primary goal is to improve the quality and safety of patient care worldwide through standardized accreditation processes and consultations.

JCI standards are adapted to the social, economic, and legal context of each country and healthcare organization. While institutions are required to conduct internal self-assessments and engage in quality improvement cycles, these results are not directly considered in the final accreditation decision. The accreditation focuses on three primary domains (Figure 3): reducing risks posed by the facility environment, improving key processes of care, and safeguarding patient dignity and rights [37].

A distinguishing feature of JCI accreditation is the opportunity for international benchmarking. Institutions accredited by JCI can compare their performance with peer organizations globally, fostering the adoption of evidence-based practices adaptable across different regulatory systems. JCI accreditation affirms that a healthcare center has achieved a level of performance aligned with international best practices [37].

### 6.3. Canadian Healthcare Council

The Canadian Healthcare Council is an independent accrediting organization with a presence in five countries that is supported by healthcare institutions across Europe, the Americas, and Canada. Its accreditation framework includes four developmental stages [46]: (i) proactive engagement in building a culture of patient safety, (ii) the establishment of essential quality infrastructure, (iii) the standardization and implementation of care processes, and (iv) the consolidation of the quality infrastructure.

What differentiates this model is that its programs are designed by healthcare professionals who have firsthand experience in implementing national and international quality certification programs. The Canadian Healthcare Council’s model is based on a maturity-level approach that provides institutions with a structured roadmap to progressively improve care quality and develop a strong culture of patient safety [46].

Through this process, the council accredits healthcare institutions that meet established standards in quality and safety. This model not only aims to improve patient outcomes but also enhances operational efficiency and process optimization across healthcare systems both within and beyond Canada.

## 7. Discussion

Defining “quality of care” remains a complex endeavor due to the multifaceted nature of healthcare systems and the diverse socioeconomic, cultural, and structural contexts in which they operate. While each system interprets and applies the concept of quality based on its unique needs, constraints, and goals, there is widespread agreement that the patient must remain at the center of any quality framework. In this article, we have proposed a conceptual synthesis that integrates insights from internationally recognized quality models and frameworks with practical clinical applications, aiming to foster improved evaluation strategies and evidence-based decision-making. Despite the absence of a universally accepted model for assessing healthcare quality, the literature supports the utility of various conceptual tools—such as the WHO model, SERVQUAL, and the Importance–Performance Analysis (IPA) framework—that contribute to understanding and enhancing service delivery. These tools, while differing in scope and emphasis, offer valuable lenses through which patient expectations and experiences can be analyzed. The SERVQUAL model, for example, captures the perceived service gap across five functional dimensions (tangibility, reliability, responsiveness, assurance, and empathy), while IPA enables the prioritization of resources and interventions based on performance–importance mapping.

By integrating these tools, a synergistic evaluation approach can be achieved. SERVQUAL provides granular insight into the discrepancies between patient expectations and perceived service quality, while IPA contextualizes these discrepancies within a matrix that helps healthcare leaders distinguish between areas that require urgent attention and those with lower impact. This dual application addresses some of the blind spots that may arise when using either tool independently, such as a lack of prioritization in SERVQUAL or the absence of root cause identification in IPA. Nevertheless, significant limitations remain. As highlighted in our review, these tools were primarily developed within specific populations or service industries and may not fully capture the nuanced dynamics of clinical outcomes or organizational constraints across all healthcare settings. Additionally, cultural biases, linguistic challenges, and the influence of patients’ socioeconomic status can distort both expectation and perception scores. To mitigate these issues, it is crucial to adapt tools through rigorous cultural validation, improve training for data collectors, and incorporate qualitative feedback alongside quantitative scoring.

The integrative conceptual model introduced in this review contributes a novel framework that positions quality assessment within a concentric structure, balancing institutional accountability and patient-centered care (Figure 4). Unlike traditional models that isolate either technical or functional aspects, our model encourages alignment among system infrastructure, provider competencies, and patient experiences. It highlights the interplay among structure, process, and outcome while acknowledging the importance of balance as a fourth pillar. This balance encompasses the dynamic negotiation between service quality dimensions and contextual limitations, ensuring that quality improvement is both feasible and impactful. Applied across diverse health systems, this model may help bridge the gap between strategic evaluation and operational implementation. It is particularly suitable for resource-limited or hybrid healthcare systems, in which existing models often fail to accommodate fragmented infrastructures. Furthermore, by enabling the visualization of overlapping responsibilities and patient–provider interactions, it supports more adaptive and responsive healthcare management practices.

This figure presents a comprehensive, multi-layered conceptual model that synthesizes the key components involved in improving healthcare quality, as discussed throughout this article. At its core lies the central objective: “Improving Quality in Healthcare Systems”, which is surrounded by two major domains—dimensions and variables—represented in a concentric structure. The dimensions domain includes fundamental aspects of care quality, such as service delivery, social accountability, assurance, responsiveness, tangibility, reliability, and empathy. These elements represent the qualitative attributes that define patient-centered, efficient, and equitable healthcare. The variables domain encompasses three distinct yet interconnected categories: variables dependent on personal attention (e.g., staff interaction and training), infrastructure-dependent variables (e.g., equipment and facility readiness), and patient-dependent variables (e.g., adherence, perception, and health literacy). These components reflect the dynamic inputs that influence how quality is implemented and experienced. Encircling these domains is a third layer that highlights the mechanisms for external evaluation: certifications and accreditations. These processes validate that healthcare institutions adhere to recognized quality standards and continuously strive for improvement. Together, the model encapsulates the interplay between internal organizational factors and external evaluative frameworks, offering a unified view of the path toward enhanced healthcare quality. It serves as a visual summary of the article’s conceptual foundation and emphasizes the need for an integrated, systems-based approach to quality improvement.

While each conceptual model offers valuable insights into the assessment of healthcare quality, their applicability varies depending on the context. The Donabedian model, although foundational, tends to oversimplify patient-centered dimensions by primarily focusing on structural and process indicators. In contrast, the SERVQUAL model captures subjective experiences and expectations, yet may lack alignment with clinical outcomes. The person-centered care (PCC) framework represents a more holistic and modern paradigm, integrating patients’ values and shared decision-making into care processes. Nevertheless, implementation challenges persist across models. For instance, SERVQUAL’s dependence on patient perception surveys may introduce response bias, while PCC frameworks often require systemic changes in provider training and institutional culture. Furthermore, none of the models individually account for macro-level determinants, such as policy or resource distribution, which are particularly relevant in low- and middle-income countries.

This synthesis highlights the importance of adopting a hybrid or context-adapted model that draws from the strengths of each framework while mitigating their limitations. For healthcare systems aiming to improve quality in a sustainable manner, integrating quantitative performance metrics with patient-reported experience measures (PREMs) could bridge the gap between technical efficiency and humanized care.

## 8. Perspectives, Implications, and Conclusions

### 8.1. Implications

The proposed model has specific implications for key stakeholders:For researchers: It provides a conceptual base for developing new tools and indicators that capture both human and systemic variables in quality assessment. This dual emphasis encourages more nuanced and inclusive measurements of healthcare quality.For healthcare administrators: It serves as a practical guide to identifying institutional weaknesses, aligning resource allocation with contextual needs, and designing targeted, sustainable quality improvement strategies.For policymakers: The model offers a flexible and comprehensive structure that can support the development of regulatory frameworks and inform health policy formulation, especially within pluralistic or decentralized healthcare systems.Its adaptability to different health system contexts enhances its value for both high-income and resource-constrained settings, fostering more equitable and responsive strategies for quality improvement initiatives.

### 8.2. Limitations

Despite its conceptual contributions, this narrative literature review is not without limitations. As it did not include a formal critical appraisal process, selection bias may have been introduced by relying on conceptual relevance over standardized quality scoring. Moreover, much of the included literature originates from high-income countries, which may constrain the generalizability of findings to low-resource settings. The narrative approach, while flexible, also limits reproducibility. These limitations underscore the need for empirical validation of the proposed model across varied healthcare environments.

### 8.3. Recommendations

Based on the evidence reviewed, we present the following recommendations:Healthcare institutions should adopt hybrid quality assessment approaches that integrate structural/process indicators with patient-reported experience measures (PREMs).Policymakers and healthcare administrators should adapt quality frameworks to local realities, prioritizing flexibility and cultural relevance, especially in low- and middle-income countries.Training programs for healthcare professionals should incorporate core competencies in patient-centered care and quality improvement to support practical implementation.Further narrative and mixed-methods research is encouraged to explore quality assessment in underrepresented healthcare systems and empirically validate the proposed model.

### 8.4. Final Reflection

This narrative review highlights the complexity of assessing healthcare quality, emphasizing the importance of context-sensitive, multidimensional approaches. Improving healthcare systems requires models that are both integrated and adaptive, capturing the balance between objective indicators and subjective human experiences. The ultimate goal is to ensure that quality assessment frameworks are grounded in institutional realities while remaining responsive to patient needs. Placing the patient at the heart of evaluation—without neglecting systemic intricacies—will be key to achieving high-quality, equitable, and sustainable care worldwide.

## Figures and Tables

**Figure 1 healthcare-13-01714-f001:**
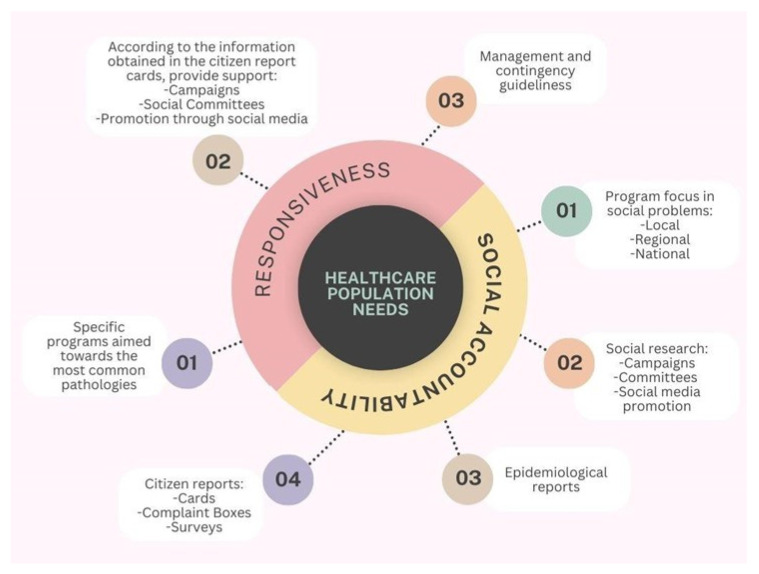
Interrelationship between responsiveness and social accountability in addressing healthcare population needs.

**Figure 2 healthcare-13-01714-f002:**
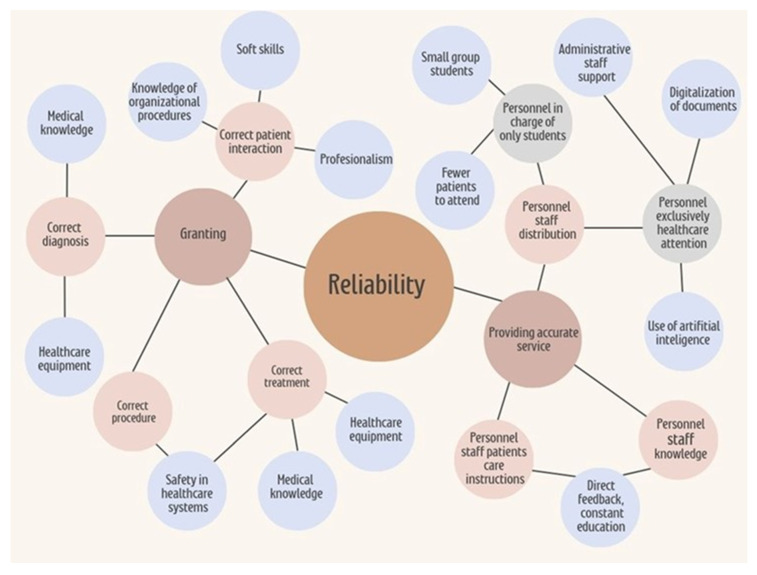
Structural and human components involved in healthcare service reliability.

**Figure 3 healthcare-13-01714-f003:**
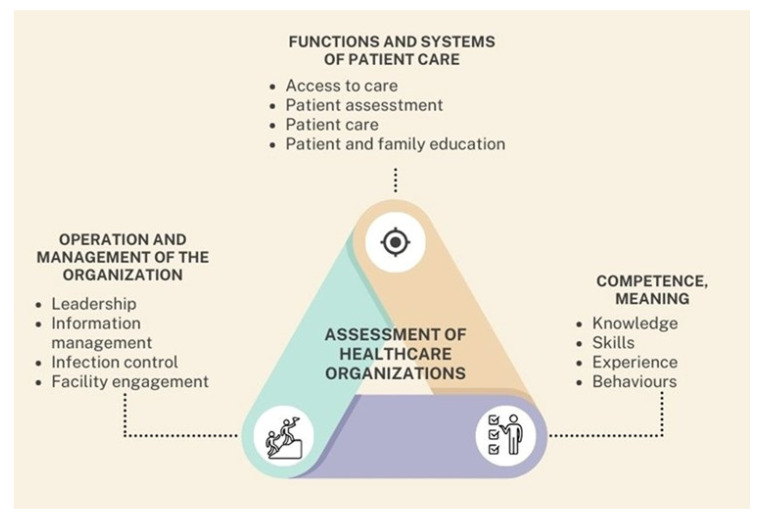
Triangular framework for the assessment of healthcare organizations.

**Figure 4 healthcare-13-01714-f004:**
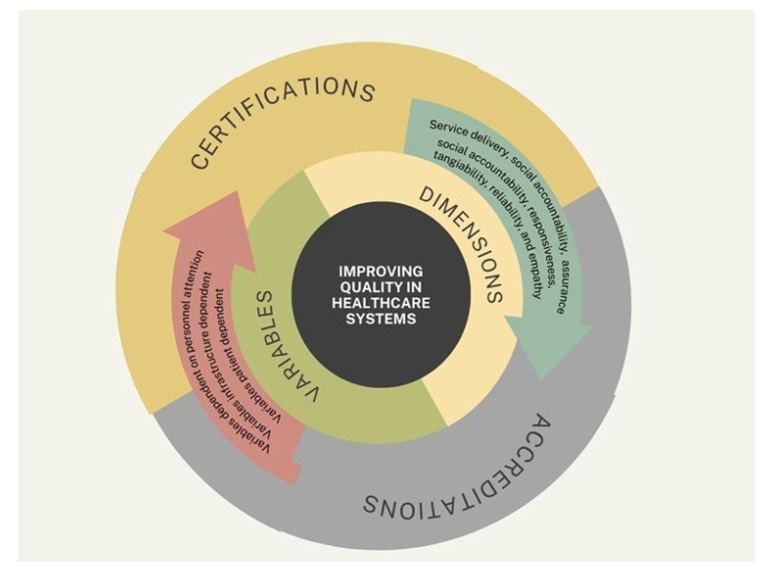
Integrative model for improving quality in healthcare systems.

**Table 1 healthcare-13-01714-t001:** Conceptual design of research in the literature.

**Search Items**		
**Initial terms** Quality of care Quality healthcare	**Topical research** Quality healthcare improvement, quality models, service quality, healthcare systems	**AND** Strategies for improvement, certifications, and accreditations
**Academic databases** Google Scholar, PubMed, Cochrane Library, Scopus, Clarivate, and Clinical Key		
**Types of research** Systematic reviews, meta-analyses, randomized controlled trials (RCT) *, observational studies, literature reviews, and discussion articles.		
**Language** English		
**Publication year limits** 2000–2024 **		

**Note:** * Studies were included to explore administrative methodologies; ** Exceptions were made for relevant older studies or when recent data were lacking.

**Table 2 healthcare-13-01714-t002:** Dimensions of quality in healthcare systems.

Dimensions to Take into Account to Improve Quality in Healthcare Systems		
** *Patient-dependent variables* **	** *Infrastructure-dependent variables* **	** *Variables dependent on personnel attention* **
Language, expectation, and perception	Tangibility, modern structural facilities, staff appearances, healthcare equipment, service organization, service delivery, efficiency, accessibility, acceptability, and safety	Reliability, responsibility, assurance, empathy, social accountability, responsiveness, effectiveness, equity, and safety

Note: None of the variables is hierarchically superior to the others; they depend on the needs of the healthcare institution’s population. The concept of each term is explained in detail in the text.

**Table 3 healthcare-13-01714-t003:** Comparison of quality assessment models.

Model	Strengths	Possible Outcomes	Context of Application	Potential Biases	Limitations	Mitigation Strategies
**WHO Quality Framework**	Comprehensive, system-level scope; aligns with global health goals; useful for policymaking and benchmarking.	Provides high-level insights into systemic strengths and weaknesses; enables benchmarking against international standards; identifies gaps in accessibility, equity, and safety.	Health system evaluations, accreditation frameworks, national reforms, and public health programs.	May favor systemic indicators over patient-centered data; bias from administrative source interpretations.	Broad and not specific for micro-level analysis; may require adaptation for local application; lacks direct patient perception.	Supplement with patient-reported outcome measures and experience surveys; disaggregate data to detect local disparities; involve stakeholders in adapting global indicators.
**SERVQUAL**	Captures patient perceptions; identifies service quality gaps; simple structure; adaptable to many healthcare settings.	Identifies specific service dimensions where patient expectations are not met; reveals perception gaps across functional quality indicators; supports targeted quality improvement plans.	Service delivery evaluations, patient satisfaction surveys, and quality improvement in clinical settings.	Recall bias, cultural bias in translation, and non-response bias in patient surveys.	Sensitive to cultural context and translation issues; lacks objective outcome measures; susceptible to response bias and small sample size limitations.	Translate and culturally adapt survey instruments; train data collectors to ensure consistency; use complementary qualitative methods (e.g., focus groups) to validate findings.
**Importance-Performance Analysis (IPA)**	Helps prioritize resource allocation; intuitive matrix visualization; complements perception-based models like SERVQUAL.	Maps service attributes according to priority and performance; helps institutions distinguish between critical and non-critical service areas; supports efficient resource reallocation.	Operational decision-making, resource prioritization in hospitals and clinics, especially useful in combination with SERVQUAL.	Confirmation bias in importance ratings; variability in interpretation; cognitive overload for respondents.	Relies heavily on subjective perceptions; limited clinical outcome insight; results depend on consistency and interpretation of importance ratings.	Clarify rating instructions for respondents; pair IPA with outcome-based metrics; perform sensitivity analyses to test robustness of importance–performance alignment.
**SERVQUAL + IPA (Combined Application)**	Provides dual insight into perception and prioritization; enables targeted, patient-centered quality improvement; enhances strategic decision-making.	Identification of mismatches between perceived service importance and performance; prioritization of service attributes needing improvement; clearer insights for tailored intervention planning.	Ideal for institutional quality diagnostics, hospital-level quality benchmarking, targeted service redesign, and prioritization of patient-centered actions.	Anchoring bias; misclassification in quadrant interpretation; omission of underreported service dimensions.	Cumulative biases from both tools; risk of overemphasis on perception over clinical quality; may neglect structural/systemic constraints.	Pre-testing instruments for linguistic/cultural fit; triangulation with objective indicators; training staff on survey delivery; ensuring appropriate sample sizes; combining quantitative data with patient interviews.

Note: In this table, we synthesize the comparative strengths, limitations, and contextual applications of the WHO framework, SERVQUAL, and IPA in a structured table to provide an actionable reference for healthcare administrators and policymakers.

**Table 4 healthcare-13-01714-t004:** Certification and accreditation differences.

Certification and Accreditation Differences	
Certification	Accreditation
Defines, standardizes, and implements specific processes	Assesses the integration of the operational institutional entity
Individual functions	Global functions
Adherence to guidelines (determined by specific committees)	Overall performance of the organization
Examples: Laboratory certification, technology and devices certification, professional competence, International Organization for Standardization 9001	Examples: Joint Commission International, Canadian Healthcare Council

## Data Availability

No new data were created or analyzed in this study.

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
