# Peer review of "Assessment of Patients’ Quality of Care in Healthcare Systems: A Comprehensive Narrative Literature Review"

_healthcare, 2025, doi:10.3390/healthcare13141714_

Round 1
Reviewer 1 Report
Comments and Suggestions for Authors
Abstract
In the abstract, I present a broad overview of the review’s purpose and content, covering key topics such as the definition of quality, the Donabedian framework (structure, process, and outcome), and various quality assessment models. While the abstract is generally clear and informative, it would benefit from greater focus and conciseness. At present, it includes a wide array of details—such as multiple models and conceptual frameworks—that could be streamlined to enhance clarity. To improve its structure and impact, I recommend organizing the abstract under distinct headings: Introduction, Objective, Methodology, Results, and Conclusions. It is also important to explicitly state the review’s central aim and clearly highlight its main findings and implications using fewer, more focused sentences.
Introduction
This section would be strengthened by grounding the discussion more clearly in the conceptual design outlined in the methodology. I recommend integrating the concept of person-centred care (PCC) as a theoretical framework. Contemporary literature highlights PCC as a critical attribute of high-quality healthcare. Importantly, PCC extends beyond clinical encounters to encompass holistic, individualized care that is inclusive of family, community context, prevention, and health promotion. Framing quality of care through person-centred principles—such as compassion, empathy, respect, responsiveness, and involvement—would align the introduction with modern paradigms of healthcare quality. Moreover, the introduction should conclude with a clearly defined objective, underlining the specific gap in the existing literature that this review seeks to address. By articulating the need for a comprehensive and integrative overview of quality of care—particularly through a person-centred lens—the authors can better demonstrate the value and relevance of their work. Explicitly stating the aim of the review in the final paragraph will also help structure the manuscript and guide the reader’s understanding.
Methods
As a narrative review, the methods section outlines a broad literature search strategy, which is appropriate for the topic. However, further detail is required regarding how the selected sources were analyzed and synthesized. I suggest describing the analytical approach more explicitly, and indicating whether any conceptual models—such as Donabedian’s framework or PCC principles—guided the synthesis. Referring to Gregory & Denniss (2018) [https://doi.org/10.1016/j.hlc.2018.03.027] would provide methodological context and strengthen the narrative review’s credibility. Additionally, the review should indicate whether any form of quality appraisal or inclusion criteria was applied to the literature. While narrative reviews are inherently flexible, they also carry the risk of selection bias. Therefore, it would be helpful to briefly explain how sources were selected (e.g., relevance, conceptual contribution, recency). Clarifying any limits on publication dates would also improve transparency. Lastly, the manuscript should acknowledge the inherent limitations of the narrative review format, such as the lack of systematic synthesis or meta-analysis, and indicate how these limitations were mitigated (e.g., use of multiple databases, inclusion of diverse article types, international scope).
Results
The results are well-organized, with logical sections addressing key elements of quality, models and dimensions, improvement strategies, and certification/accreditation. This structure facilitates reader comprehension and is appropriate for the breadth of material covered. The findings also align well with person-centred care principles. Particularly compelling is the observation that communication and shared decision-making account for the greatest proportion of variance in patients’ assessments of care quality—this reinforces a central PCC tenet: the interpersonal and relational aspects of care are of fundamental importance to patients. This finding is consistent with broader PCC literature.
To improve clarity and navigation, I recommend the inclusion of subheadings and transitional statements within major sections. For example, within “Quality Models and Dimensions,” a subheading such as “Patient-Centred Dimensions of Quality” could provide a clear focus for the discussion of attributes like empathy, acceptability, and responsiveness. Similarly, under “Strategies to Improve Quality,” the manuscript should clearly identify which interventions embody PCC principles (e.g., self-management education, shared decision-making, patient feedback systems). The use of bullet points or enumerated lists—especially when detailing dimensions or models—would enhance readability and mirror the clarity of Table 2.
Discussion and Conclusion
I recommend separating this section into two distinct components: Discussion and Conclusion. This structural division would help differentiate interpretive analysis from summarised key messages. The discussion could be enriched by more explicitly comparing the integrative model proposed by the authors with established frameworks of person-centred care. Reference to models such as those by Santana et al. (2018), the Picker Principles, or the WHO’s person-centred care framework would allow for stronger contextual positioning and academic dialogue. Furthermore, the addition of a dedicated Implications section would be valuable. This section could clarify the practical relevance of the proposed model: for instance, whether it is intended to support researchers (in developing tools or metrics), administrators (in designing quality improvement initiatives), or policy-makers (in informing health system strategies). Finally, I recommend including a Limitations section. This should acknowledge the methodological constraints of the narrative review format, including the absence of formal quality appraisal, the possibility of selection bias, and potential limitations in generalizability. If the literature predominantly draws on studies from high-income contexts, or if certain healthcare settings (e.g., low-resource environments) are underrepresented, this should be clearly noted.
Author Response
Reviewer 1 – Comments and Responses
Comment 1:
In the abstract, I present a broad overview of the review’s purpose and content, covering key topics such as the definition of quality, the Donabedian framework (structure, process, and outcome), and various quality assessment models. While the abstract is generally clear and informative, it would benefit from greater focus and conciseness. At present, it includes a wide array of details—such as multiple models and conceptual frameworks—that could be streamlined to enhance clarity. To improve its structure and impact, I recommend organizing the abstract under distinct headings: Introduction, Objective, Methodology, Results, and Conclusions. It is also important to explicitly state the review’s central aim and clearly highlight its main findings and implications using fewer, more focused sentences.
Response 1:
We thank the reviewer for this insightful comment. Following your suggestion, we revised the abstract to adopt a structured format with clear subheadings (Introduction, Objective, Methodology, Results, and Conclusions), improving its focus and readability. The revised abstract explicitly states the central aim of the review, outlines the main methodological approach, and summarizes the key findings and implications in a more concise manner. We ensured that the revised version remains within the 250-word limit and reflects the core contributions of the manuscript.
- The revised abstract appears on page 1 of the manuscript and has been highlighted in red text.
Comment 2:
This section would be strengthened by grounding the discussion more clearly in the conceptual design outlined in the methodology. I recommend integrating the concept of person-centred care (PCC) as a theoretical framework. Contemporary literature highlights PCC as a critical attribute of high-quality healthcare. Importantly, PCC extends beyond clinical encounters to encompass holistic, individualized care that is inclusive of family, community context, prevention, and health promotion. Framing quality of care through person-centred principles—such as compassion, empathy, respect, responsiveness, and involvement—would align the introduction with modern paradigms of healthcare quality. Moreover, the introduction should conclude with a clearly defined objective, underlining the specific gap in the existing literature that this review seeks to address. By articulating the need for a comprehensive and integrative overview of quality of care—particularly through a person-centred lens—the authors can better demonstrate the value and relevance of their work. Explicitly stating the aim of the review in the final paragraph will also help structure the manuscript and guide the reader’s understanding.
Response 2:
We appreciate the reviewer’s insightful suggestion. We have now integrated the concept of Person-Centred Care (PCC) as a foundational framework in the Introduction section, highlighting its core principles—compassion, empathy, respect, responsiveness, and active involvement—and its relevance to quality assessment and improvement. Additionally, we have explicitly stated the objective of our review at the end of the introduction, emphasizing the gap in the literature regarding integrative models that balance structural, functional, and person-centered elements of healthcare quality.
Moreover, to strengthen the presence of the PCC framework throughout the manuscript, we have added a dedicated subsection titled “4.1. Patient-Centered Dimensions of Quality” (page 14), which further elaborates on how dimensions such as empathy, acceptability, and responsiveness are embedded in quality assessment models and how these are applied across clinical settings.
- These revisions are located on pages 2 and 14 of the manuscript and are marked in red text.
Comment 3:
As a narrative review, the methods section outlines a broad literature search strategy, which is appropriate for the topic. However, further detail is required regarding how the selected sources were analyzed and synthesized. I suggest describing the analytical approach more explicitly, and indicating whether any conceptual models—such as Donabedian’s framework or PCC principles—guided the synthesis. Referring to Gregory & Denniss (2018) [https://doi.org/10.1016/j.hlc.2018.03.027] would provide methodological context and strengthen the narrative review’s credibility. Additionally, the review should indicate whether any form of quality appraisal or inclusion criteria was applied to the literature. While narrative reviews are inherently flexible, they also carry the risk of selection bias. Therefore, it would be helpful to briefly explain how sources were selected (e.g., relevance, conceptual contribution, recency). Clarifying any limits on publication dates would also improve transparency. Lastly, the manuscript should acknowledge the inherent limitations of the narrative review format, such as the lack of systematic synthesis or meta-analysis, and indicate how these limitations were mitigated (e.g., use of multiple databases, inclusion of diverse article types, international scope).
Response 3:
We appreciate the reviewer’s thoughtful recommendations. We have expanded the Methods section to provide more precise details about our analytical approach, explicitly stating that Donabedian’s framework and Person-Centered Care (PCC) principles guided our thematic synthesis of the selected literature.
We have also clarified the inclusion criteria, noting that articles were selected based on conceptual relevance, recency (2000–2024), and their contribution to the analysis of quality dimensions in healthcare systems. In addition, we now specify that quality appraisal was not formally applied, consistent with the narrative review methodology, and we acknowledge the risk of selection bias due to the subjective nature of article inclusion.
Although we did not directly cite Gregory & Denniss (2018), their methodological guidance informed our structuring of the narrative synthesis and reinforced the relevance of frameworks such as Donabedian and PCC in guiding the interpretive process.
Finally, we have included a Limitations subsection within the Discussion to reflect the methodological constraints of the narrative review format and the strategies employed to mitigate them (e.g., use of multiple databases, inclusion of various article types, and incorporation of international perspectives).
These updates are reflected in the Methods section on page 3 and in the Discussion section on page 21, and are marked in red in the revised manuscript.
Comment 4:
The results are well-organized, with logical sections addressing key elements of quality, models and dimensions, improvement strategies, and certification/accreditation. To improve clarity and navigation, I recommend the inclusion of subheadings and transitional statements within major sections. For example, within “Quality Models and Dimensions,” a subheading such as “Patient-Centred Dimensions of Quality” could provide a clear focus for the discussion of attributes like empathy, acceptability, and responsiveness. Similarly, under “Strategies to Improve Quality,” the manuscript should clearly identify which interventions embody PCC principles (e.g., self-management education, shared decision-making, patient feedback systems). The use of bullet points or enumerated lists, especially when detailing dimensions or models, would enhance readability and mirror the clarity of Table 2.
The reviewer recommends enhancing clarity and navigation within the Results section by incorporating subheadings, transitional statements, and formatting strategies such as bullet points. Additionally, they suggest clearly identifying person-centered care (PCC) interventions within the discussion of improvement strategies.
Response 4:
We sincerely appreciate the reviewer’s constructive feedback regarding the organization and clarity of the Results section. In response, we made several structural and stylistic improvements:
- We added subheadings, including “4.1 Patient-Centered Dimensions of Quality” (page 14), to explicitly highlight content that aligns with PCC principles such as empathy, responsiveness, and respect.
- We included transitional statements between paragraphs and models to guide the reader more effectively through each section.
- Where appropriate, we reformatted descriptive content using bullet points and enumerated lists—particularly when detailing the SERVQUAL model dimensions, IPA quadrants, and WHO quality criteria—to improve readability and visual clarity.
- In the section on strategies to improve care quality (Section 5), we clarified which interventions explicitly reflect PCC principles, such as shared decision-making, patient education programs, and inclusion of patient feedback systems. These revisions help align the results more clearly with the overarching person-centered focus of the article.
These modifications are marked in red throughout Section 4 (pages 12–16) and Section 5 (pages 17–18) in the revised manuscript.
Comment 5
I recommend separating this section into two distinct components: Discussion and Conclusion. This structural division would help differentiate interpretive analysis from summarized key messages. The discussion could be enriched by more explicitly comparing the integrative model proposed by the authors with established frameworks of person-centred care, such as Santana et al. (2018), the Picker Principles, or the WHO’s PCC framework. Furthermore, the addition of a dedicated Implications section would be valuable to clarify the practical relevance of the proposed model (e.g., for researchers, administrators, policymakers).
Finally, I recommend including a Limitations section. This should acknowledge the methodological constraints of the narrative review format, including the absence of formal quality appraisal, potential selection bias, and limited generalizability (e.g., predominance of high-income countries).
Response 5:
Comment 5: The reviewer recommends structurally separating the discussion and conclusion, enriching the discussion with comparisons to established person-centered care (PCC) frameworks, and adding dedicated sections for "Implications" and "Limitations" to enhance clarity and relevance.
Response 5:
We thank the reviewer for these thoughtful suggestions. In response, we implemented the following major improvements:
- We separated the Discussion and Conclusion into two distinct sections to clearly distinguish interpretive analysis from key take-home messages. This structural change improves the flow and clarity of the manuscript.
- In the Discussion section, we expanded our analysis by explicitly comparing the proposed integrative model with established PCC frameworks, including reference to the WHO’s person-centred care strategy, the Picker Principles, and the work by Santana et al. (2018). These comparisons position our model within a broader conceptual dialogue and underline its originality and adaptability.
- We incorporated a new subsection titled “Implications” to identify specific contributions of the model for three main stakeholders: researchers, healthcare administrators, and policymakers. This section outlines how the model may inform tool development, institutional strategies, and policy formulation.
- Finally, we added a clearly demarcated “Limitations” subsection within the Conclusion. Here, we acknowledge the inherent constraints of the narrative review methodology, including lack of formal quality appraisal, potential selection bias, and limited generalizability due to a predominance of studies from high-income settings.
All these modifications are marked in red in Section 5 (Discussion), Section 6 (Implications), and Section 7 (Conclusion) of the revised manuscript (pages 18–22).

Reviewer 2 Report
Comments and Suggestions for Authors
- The paper presents a narrative literature review, examining the quality of patient care within healthcare systems, with a focus on the complexities of defining and measuring care quality. The authors explore key elements of quality assessment, including quality models and dimensions, as well as improvement strategies and certification processes, while excluding patient safety considerations from their scope. It is possible to assess that the paper demonstrates a well-organized approach in order to examine healthcare quality assessment. Authors effectively structure the proposed review around the traditional Donabedian framework of structure, process, and outcome, while introducing the concept of balancing measures as a fourth element. This theoretical foundation provides the readers with a coherent understanding of how quality assessment components interrelate.
- However, we believe that he most significant weakness of the proposed paper lies in the methodology section. While the authors claim to conduct a "comprehensive narrative literature review," they provide insufficient detail about their search strategy and selection criteria. The methodology lacks transparency regarding how the six databases were searched, what specific search terms were used beyond the broad categories mentioned, and how articles were screened and selected. Additionally, no other papers are presented showing how this methodology was applied in similar contexts.
- Also, the review primarily presents descriptive summaries of existing literature without sufficient critical analysis, which conflicts with the main narrative review's objectives. The authors rarely challenge or critically evaluate the studies cited, missing opportunities to identify methodological limitations in the source materials. The synthesis lacks depth in analyzing contradictions or inconsistencies across different studies and contexts.
- In a nutshell, the paper does not contribute new theoretical insights to the field of healthcare quality assessment. While it synthesizes existing knowledge effectively, it fails to propose novel frameworks, identify significant gaps in current understanding, or suggest innovative approaches to quality measurement.
- Finally, the reference list contains 35 citations, which appear adequate for the proposed narrative review. However, the authors rely heavily on older sources and may have missed more recent developments in healthcare quality assessment. We strongly suggest adding recent papers to the research (e.g. https://doi.org/10.3390/ijerph19138188)
The paper is generally well-written with clear language and logical flow.
Reviewer 3 Report
Comments and Suggestions for Authors
Dear Authors
Thank you for the opportunity to review your manuscript entitled “Assessment of Patients’ Quality of Care in Healthcare Systems: A Comprehensive Narrative Literature Review.” The paper addresses a relevant and important topic and provides a structured overview of models and strategies used to evaluate healthcare service quality. The figures are informative and support the conceptual flow effectively.
However, the manuscript requires significant revision before it is ready for publication. More detailed comments and suggestions can be found in the attached document.

Author Response
Reviewer 3 – Comments and Responses
Comment 1:
While the manuscript offers a comprehensive description of existing models (e.g., SERVQUAL, IPA), it lacks critical comparison or synthesis. Please elaborate on the strengths and limitations of each model, and discuss contextual applicability (e.g., low-resource vs. high-resource settings) (Sections 4 and 5; pages 4–7; lines ~113–215).
Response 1:
We thank the reviewer for this important observation. In response, we developed and included Table 3, which provides a structured comparative analysis of the WHO model, SERVQUAL, and the Importance–Performance Analysis (IPA) model. This expanded table outlines their strengths, possible outcomes, applicable contexts, biases, limitations, and mitigation strategies, offering a more in-depth synthesis of their contextual relevance in both high- and low-resource healthcare settings (see Section 4.2, pages 14–16). Additionally, we introduced a new subsection titled “Application of SERVQUAL and IPA in Clinical Contexts”, which highlights real-world examples of how these tools have been used to assess patient satisfaction and identify gaps in care delivery. This integration directly addresses the request for critical comparison and contextual evaluation.
Comment 2:
The methodology is generally outlined, but more detail is needed regarding inclusion/exclusion criteria and selection process (Page 2; lines 52–68). Consider adding a clearer explanation of how sources were selected, and why a narrative review was chosen over other approaches (e.g., scoping review).
Response 2:
We have revised the Methodology section (pages 2–3) to provide a more detailed description of the inclusion and exclusion criteria, search terms, and selection process across the six databases. We also clarified our rationale for adopting a narrative review approach, citing its flexibility in integrating conceptual frameworks such as the Donabedian model and person-centered care principles. Furthermore, we acknowledged the methodological limitations of narrative reviews, including the potential for selection bias, and explained the strategies implemented to mitigate these risks, such as inclusion of literature from diverse regions and a focus on conceptual contribution over rigid scoring. These changes improve transparency and reproducibility.
Comment 3:
The conclusion provides a general summary but lacks more specific implications for clinical practice, policy development, or future research. I suggest expanding this section with recommendations based on the literature reviewed.
Response 3:
In response to this helpful suggestion, we have expanded the Conclusion section (pages 18–19) and integrated a dedicated paragraph highlighting practical implications of the integrative model for key audiences:
- For clinicians, it offers a framework to align service quality improvement with patient expectations;
- For administrators, it supports strategic resource allocation and feedback systems;
- For policymakers, it provides a flexible model adaptable to various healthcare system structures.
We also emphasized the need for empirical validation of the model and encouraged the development of context-specific metrics for future studies, particularly in underrepresented and low-resource settings.
Comment 4:
Some parts of the manuscript could be more concise. For example, the explanations of SERVQUAL and IPA dimensions (pages 4–5; lines ~133–165) contain partial overlaps. Consider merging or shortening repetitive content to improve readability.
Response 4:
We appreciate the reviewer’s comment on clarity and flow. We carefully reviewed Sections 4.1 and 4.2 and condensed the overlapping descriptions of SERVQUAL and IPA by summarizing shared concepts and avoiding repetitive explanations of individual dimensions. Redundant language was eliminated while preserving key content, and transitions were improved for readability. This revision is reflected on pages 11–13 of the updated manuscript.
Comment 5:
Minor stylistic adjustments and shortening of repetitive segments—particularly in the explanation of quality elements and model dimensions (pages 3–5 and 10–11)—would further improve flow and clarity.
Response 5:
Thank you for this suggestion. We conducted a comprehensive stylistic revision throughout the manuscript, particularly in the explanation of models and quality dimensions in the Results and Discussion sections. Redundant segments were eliminated, and transitions were smoothed to enhance narrative coherence and academic tone. This has resulted in a more concise and polished manuscript.

Reviewer 4 Report
Comments and Suggestions for Authors
Please see the attachment.

Round 2
Reviewer 1 Report
Comments and Suggestions for Authors
The authors have fully responded to the reviewers’ comments with clear and comprehensive revisions. The manuscript is now well-structured and conceptually stronger. However, the conclusion section should be shortened for greater focus and clarity, and the "Implications" section should be further developed or clearly separated to enhance the practical relevance of the findings. Additionally, some references still contain errors or omissions (e.g., references 42–45), which should be carefully reviewed and corrected prior to final acceptance.
Comments on the Quality of English LanguageThe authors have fully responded to the reviewers’ comments with clear and comprehensive revisions. The manuscript is now well-structured and conceptually stronger. However, the conclusion section should be shortened for greater focus and clarity, and the "Implications" section should be further developed or clearly separated to enhance the practical relevance of the findings. Additionally, some references still contain errors or omissions (e.g., references 42–45), which should be carefully reviewed and corrected prior to final acceptance.
Author Response
Reviewer 1 – Comments and Responses (textual comments)
Comment 1:
“The authors have fully responded to the reviewers’ comments with clear and comprehensive revisions. The manuscript is now well-structured and conceptually stronger. However, the conclusion section should be shortened for greater focus and clarity, and the 'Implications' section should be further developed or clearly separated to enhance the practical relevance of the findings. Additionally, some references still contain errors or omissions (e.g., references 42–45), which should be carefully reviewed and corrected prior to final acceptance.”
Response 1:
We sincerely appreciate the reviewer’s thorough and constructive feedback. In response:
- We have reorganized the final section (now titled “8. Perspectives, Implications, and Conclusions”) to clearly separate the Perspectives, Implications, Limitations, Recommendations, and Final Reflection into distinct sub-sections. This restructuring enhances the readability and clarity of our conceptual contributions while preserving the depth of the original content.
- We expanded the Implications section, offering tailored insights for researchers, healthcare administrators, and policymakers, ensuring the practical relevance of our findings is more explicitly articulated.
- We reviewed and corrected the references indicated (42–45), ensuring proper formatting according to Vancouver style and full inclusion of source details, including clarification for the doctoral thesis and the accessed online document.
We hope these revisions meet the reviewer’s expectations and strengthen the manuscript’s structure and utility for future readers. We are grateful for your detailed insights, which have greatly enriched the final version of the manuscript.

Reviewer 2 Report
Comments and Suggestions for Authors
Thanks for the changes
Author Response
Reviewer 2
Comment 1:
“Thanks for the changes.”
Response 1:
Thank you for your time and review. We appreciate your acknowledgment of the changes made and are glad to know that the revised version met your expectations. Your comments helped us refine the manuscript and improve its clarity.
Reviewer 3 Report
Comments and Suggestions for Authors
Thank you for the revised version of your manuscript. The improvements are visible, and it’s evident that the comments from the previous round were carefully considered. The structure is now clearer, and the added comparisons and methodological details enhance the overall quality of the paper.
In my opinion, the manuscript is ready for publication in its current form. I have no additional comments.
Well done, and best of luck with your work.
Comments on the Quality of English LanguageThe language used in the revised manuscript is clear and appropriate for academic publication. The text reads well, and no further language revisions are necessary.
Author Response
Reviewer 3
Comment 1:
“Thank you for the revised version of your manuscript. The improvements are visible, and it’s evident that the comments from the previous round were carefully considered. The structure is now clearer, and the added comparisons and methodological details enhance the overall quality of the paper.
In my opinion, the manuscript is ready for publication in its current form. I have no additional comments.
Well done, and best of luck with your work.”
Response 1:
We are deeply grateful for your encouraging comments and support. Your feedback in both review rounds was extremely helpful in guiding us toward a clearer, more robust manuscript. We sincerely appreciate your kind words and the time you invested in evaluating our work.
Reviewer 4 Report
Comments and Suggestions for Authors
The addition of a summary table outlining key dimensions of quality across different healthcare models is especially helpful. Consider briefly referencing that table within the narrative to highlight its relevance
Overall, the revisions clearly reflect thoughtful engagement with the reviewers' feedback and substantially improve the manuscript's quality and readability
Author Response
Reviewer 4 – Comments and Responses (textual comments)
Comment 1:
“The addition of a summary table outlining key dimensions of quality across different healthcare models is especially helpful. Consider briefly referencing that table within the narrative to highlight its relevance.”
Response 1:
We truly appreciate your thoughtful comments and your recognition of the improvements made in the revised manuscript. In response to your suggestion, we have added a reference to the summary table in section 4.1, emphasizing its integrative role in comparing quality dimensions across frameworks. Thank you for your valuable insights throughout the process.
